# Parenchymal Cavitations in Pulmonary Tuberculosis: Comparison between Lung Ultrasound, Chest X-ray and Computed Tomography

**DOI:** 10.3390/diagnostics14050522

**Published:** 2024-02-29

**Authors:** Diletta Cozzi, Maurizio Bartolucci, Federico Giannelli, Edoardo Cavigli, Irene Campolmi, Francesca Rinaldi, Vittorio Miele

**Affiliations:** 1Radiology Emergency Department, Careggi University Hospital, 50139 Florence, Italy; edoardocavigli@yahoo.it (E.C.); vmiele@sirm.org (V.M.); 2Department of Radiology, Azienda USL Toscana Centro, 59100 Prato, Italy; mauriziobartolucci1@gmail.com; 3Department of Radiology, Azienda USL Toscana Centro, Mugello Hospital, 50032 Borgo San Lorenzo, Italy; dr.fgiannelli@gmail.com; 4Department of Radiology, Azienda USL Toscana Centro, San Giovanni di Dio Hospital, 50143 Florence, Italy; 5Department of Infectious and Tropical Diseases, Careggi University Hospital, 50134 Florence, Italy; irenecampolmi@yahoo.it; 6Department of Infectious Diseases, Azienda Ospedaliero Universitaria Maggiore della Carità, 28100 Novara, Italy; francesca.rinaldi.222@gmail.com

**Keywords:** lung ultrasound, chest X-ray, tuberculosis, computed tomography, infection

## Abstract

This article aims to detect lung cavitations using lung ultrasound (LUS) in a cohort of patients with pulmonary tuberculosis (TB) and correlate the findings with chest computed tomography (CT) and chest X-ray (CXR) to obtain LUS diagnostic sensitivity. Patients with suspected TB were enrolled after being evaluated with CXR and chest CT. A blinded radiologist performed LUS within 3 days after admission at the Infectious Diseases Department. Finally, 82 patients were enrolled in this study. Bronchoalveolar lavage (BAL) confirmed TB in 58/82 (71%). Chest CT showed pulmonary cavitations in 38/82 (43.6%; 32 TB patients and 6 non-TB ones), LUS in 15/82 (18.3%; 11 TB patients and 4 non-TB ones) and CXR in 27/82 (33%; 23 TB patients and 4 non-TB ones). Twelve patients with multiple cavitations were detected with CT and only one with LUS. LUS sensitivity was 39.5%, specificity 100%, PPV 100% and NPV 65.7%. CXR sensitivity was 68.4% and specificity 97.8%. No false positive cases were found. LUS sensitivity was rather low, as many cavitated consolidations did not reach the pleural surface. Aerated cavitations could be detected with LUS with relative confidence, highlighting a thin air crescent sign towards the pleural surface within a hypoechoic area of consolidation, easily distinguishable from a dynamic or static air bronchogram.

## 1. Introduction

According to a recent report released by the World Health Organization (WHO) in 2023, tuberculosis (TB) is the second leading cause of death from a single infective agent (after SARS-CoV-2), accounting for more than 10 million cases and around 1.3 million deaths in the world in 2022 [1]. To stop the global TB epidemic, in 2014 WHO defined the “End TB Strategy”, underlining the need to develop diagnostic methods, as well as improve treatment and prevention strategies, to ensure earlier and correct diagnosis [2]. Worldwide, TB has a differing epidemiology: low- and middle-income countries, such as sub-Saharan African and Southeast Asian regions, are affected with the highest rates of infection (“high-burden countries”); conversely, high-income countries show lower incidence (“low-burden countries”). In high-burden TB countries, which accounted for 88% of cases worldwide, TB is still a common disease, often undiagnosed due to lack of diagnostic tools. On the other hand, in low-burden TB countries (e.g., Italy, accounting for 4000 cases in 2020), the diagnostic tools are available but the poor incidence leads to misdiagnosis and delays in treatment and contact tracing [3]. Pulmonary manifestations of tuberculosis are various and partly depend on whether the infection is primary or post-primary. Pulmonary tuberculosis frequently manifests parenchymal cavitation in 40–87% of cases, but it has to be remembered that cavitations related to TB are uncommon in primary TB itself (seen only in 10–30% of cases) [3,4]. Cavitations are detectable in most relapses and treatment failures, most cases of multi-drug resistance (MDR) and nearly all cases of extensively drug-resistant TB [4,5]. Patients with lung cavitations are contagious and have a bacterial load greater than subjects without cavitations [5,6]. Moreover, these patients usually have a higher risk of complications such as tuberculous empyema, colonization of cavities (such as aspergilloma), bronchopleural fistulas or pulmonary artery pseudoaneurysms (better known as Rasmussen aneurysms). Computed tomography (CT) is the gold standard of imaging diagnosis of TB, because of its higher sensitivity than chest X-ray (CXR) in the detection and characterization of pulmonary findings [7,8]. However, both CT and CXR have several disadvantages. The main limitations are represented by ionizing radiation exposure, poor access to high-quality radiologic equipment, limited access in rural areas and prohibitive costs for patients’ diagnosis. Nowadays, lung ultrasound (LUS) is considered a valid tool in the diagnosis of many lung pathologies and is mainly a safe, portable and cost-effective imaging modality [9,10,11,12]. With LUS, it is possible to identify an interstitial syndrome characterized by a smooth thickening of the interlobular septa (represented as an increase in the B-lines) and areas of partial alveolar fillings, the so-called “white lung” corresponding to the ground glass opacities that can be highlighted in CT [9,10]. Moreover, it is possible to detect alveolar consolidations as an area of hepatization with a dynamic air bronchogram or an area of atelectasis or pneumonia in patients with a static air bronchogram [8,11]. We previously described the role of LUS in pulmonary TB in the work “Lung ultrasound (LUS) in pulmonary tuberculosis: correlation with chest Computed Tomography and X-ray findings” [11]. We analyzed a population of patients with clinical/radiological/epidemiological suspicion of pulmonary TB who accessed the Emergency Department of our hospital between September 2017 and February 2020. Overall, LUS sensitivity in detecting TB was 80%, greater for micronodules (82%) and nodules (95%) and lower for consolidation with an air bronchogram (72%) and cavitations (33%). Among the 82 patients enrolled, 48 presented cavitations at CT (42 TB patients and 6 non-TB patients), with lower identification by chest X-ray and LUS [11].

Based on the findings of our previous study, the present work aims to evaluate the potential of lung ultrasound in recognizing lung cavitations, compared with chest CT and CXR, to describe its main characteristics and diagnostic accuracy.

## 2. Materials and Methods

### 2.1. Study Setting and Patient Selection

This prospective monocentric study was conducted in the period between September 2017 and February 2020 in our Florentine University Hospital. All patients were admitted with clinical, radiological and epidemiological suspicion of pulmonary TB. They all performed LUS examinations within three days after admission. Both CXR and CT exams were performed at admission to the Radiology Emergency Department in the same institute.

### 2.2. TB Diagnosis and Imaging

The diagnosis of pulmonary TB was based on clinical findings (dyspnea, fever, weight loss, persistent cough), computed tomography and chest X-ray scan images suggestive of pulmonary TB and bacteriological confirmation. Depending on clinical features, we collected sputum, bronchoalveolar lavage (BAL) fluid, pleural liquid or material from a transbronchial biopsy (TBNA) for microbiological analysis. All CT scans were obtained using a multidetector 128-slice CT (Brilliance 128 iCT SP, Philips Medical System, Amburg, Germany). Patients were scanned in a supine position with craniocaudal acquisition and suspended breath. Technical parameters were slice thickness 1 mm, reconstruction filter B70 lung, KV 120, Dose Right-Index 19 and Pitch 1.4. All chest X-ray scans were obtained as digital radiographs in the X-ray room (Digital Diagnost 4.1.x, Philips Medical System) or with the same portable X-ray unit (FDR Go PLUS, Fujifilm, Cernusco sul Naviglio, Italy). Only a small number of patients were scanned with two projections (postero–anterior (PA) and later lateral); most were scanned in a supine position with an antero–posterior (AP) projection, so we decided not to evaluate the lateral view for all patients. An echographic examination was performed by three thoracic expert radiologists (with expertise in thoracic ultrasounds) and two doctors in training, all blinded about the patient’s radiological examinations and clinical status; in case of disagreement, common agreement was found. The exam was conducted using an ESAOTE model (MyLab Class C Advance, Genova, Italy) with a 5–3.5 MHz convex probe or 7.5 MHz linear probe. Patients were examined both in a seated and in a supine position. Examinations were performed by taking longitudinal scans starting anteriorly from the parasternal zone and posteriorly from the paravertebral/posterior axillary lines to analyze every intercostal space. The examination of lung apexes was performed by applying the probe vertically between the clavicle and the trapezius muscle anteriorly. The whole surface of the chest was thus analyzed. The radiologist filled out a predefined form where the localization of the cavitations was outlined. Each marker was localized as apical, middle or inferior, anterior or posterior or left and right with the same approach as explained in our previous work [11]. At CXR, we were not able to evaluate anterior and posterior regions with only AP or PA projections, so we localized CXR without considering anterior and posterior regions.

### 2.3. Statistical Analysis

Data were analyzed using STATA/MP (version 14 STATA Corp., College Station, TX, USA). Epidemiological, clinical and demographic features were analyzed by adequate descriptive statistics. Continuous variables were expressed with median and interquartile range and categorical variables as proportions. Differences between groups were assayed using the Wilcoxon Ranks Test or Chi Square Test. Statistically significant *p*-values were defined as <0.005. The diagnostic accuracy of cavitation detection was assessed and the sensitivity of LUS and CXR versus CT (gold standard) was calculated.

## 3. Results

This study was approved by the Ethics Committee (ref. CEAVC 14816). A total of 82 patients with suspicion of pulmonary TB were enrolled in this study (51 males, 31 females; median age 48 years old). In 58 patients, laboratory examinations confirmed the diagnosis of pulmonary TB with a disease prevalence of 71%. The epidemiological and anamnestic features of the patients in this study are shown in Table 1; differential diagnoses in non-pulmonary TB cases are described in Table 2.

Chest CT showed pathological cavitations in 38/82 patients (46.3%), of which 32/58 had pulmonary TB (55.2%) and 6/24 non-TB (25%); 11 of 32 pulmonary TB patients (34.3%) showed cavitations in multiple areas, and the 6 non-tuberculous patients showed cavitation in a single area (upper anterior, upper posterior, middle anterior, middle posterior, inferior anterior and inferior posterior, all to the right and left, respectively) (Table 3).

LUS showed pathological cavitations in only 15/82 patients (18.3%), of which 11/58 had pulmonary TB (19%) and 4/24 non-tuberculous (16.7%); only 1 pulmonary TB patient (7%) showed cavitations in multiple areas. Only subpleural pulmonary cavitations, with direct pleural contact, were visible on ultrasound examination. The cavity’s dimension does not correlate with its ultrasound detection; the only limit is precisely whether it reaches the pleural surface or not, as otherwise it is difficult to demonstrate. The ultrasound examination was able to highlight even small centimeter-wide lesions but also larger lesions, with a depth of up to 7–8 cm using the convex probe.

Chest X-rays showed pathological cavitations in 27/82 patients (33%), of which 23/58 had pulmonary TB (40%) and 4/24 non-tuberculous (16.7%); only 3 of 32 pulmonary TB patients (9.4%) showed cavitations in multiple areas (upper, middle and inferior, all to the right and left, respectively, without evaluation of anterior or posterior areas in the AP evaluation only). Non-TB patients with cavitations have bacterial pneumonia as a definitive diagnosis in three cases, a lung tumor in two cases and, in one case, an atypical mycobacteriosis. One of the patients with positive CXR did not report cavitation in the area corresponding to the CT scan and therefore was considered a false positive. Considering CT as the gold standard of reference, we obtained the sensitivity, specificity, positive predictive values (PPVs) and negative predictive values (NPVs) for LUS and CXR. These characteristics for LUS are sensitivity 39.5%, specificity 100%, PPV 100% and NPV 65.7%. Instead, for CXR, these characteristics are sensitivity 68.4%, specificity 97.8%, PPV 96.3% and NPV 78.2%.

## 4. Discussion

The results of this study show how lung ultrasound can detect air cavitations with relative confidence by highlighting a thin air crescent convex towards the pleural surface within a hypoechoic area of consolidation, easily distinguishable from the dynamic and static aerial bronchogram (Figure 1). No false positive exams were seen in our study. The sensitivity of LUS in identifying cavitations turned out to be rather low as many cavitated consolidations do not reach the pleural surface or are located behind the scapula, a typical region of post-primary TB. On the other hand, an advantage of LUS is that during the examination it is not necessary to apply a lot of force with the ultrasound probe (both linear and convex) on the chest wall, because the pleural surface is usually located a few cm from the skin. This makes learning and performing the LUS exam even simpler and easier. As previously performed by other authors, we investigated patients with consolidation (myco-tuberculous, bacterial or neoplastic) of the tissues surrounding the cavitation, which allowed us to better highlight the crescent of air’s sickle within the hypoechoic consolidation [13]. In patients without consolidation surrounding the air cavitation, it could appear difficult to distinguish the air in the cavitation from the air in the lung [7,14]. If computed tomography or CXR (which have a significantly higher sensitivity) are not available, an ultrasound bedside assessment of the pulmonary findings may be useful. Moreover, it is possible to carry out ultrasound monitoring and follow-up of cavitations without ionizing radiation; in fact, through computed tomography and/or CXR we can understand which cavitations reach the pleural surface and could be visible with a LUS exam and, therefore, which findings can be monitored by ultrasound itself [8,14,15]. In fact, a fundamental role of LUS in patients with pulmonary cavitations is the disease follow-up during and after medical therapy.

Cavitations are not pathognomonic of TB as they can also be found in patients with bacterial pneumonia, atypical mycobacteriosis, abscesses, septic emboli, aspergillosis, granulomatosis with polyangiitis and malignant tumors [16,17,18,19,20,21,22]. With LUS, we were able to identify lung air cavitations without a close relationship with dimensions (both large and small).

Moreover, we found three essential prerequisites to identify cavitations: a large consolidated area in proximity, the area of consolidation reaching the pleural surface and the possibility of scanning the pleural surface at this level. In fact, most of the cavitations that we were unable to evaluate by LUS were surrounded by small areas of parenchymal thickening that did not reach the pleura or were located at the upper or middle posterior regions because they were hidden by the scapula [22]. With LUS, it is possible to detect the presence of a thin crescent of air with a slightly convex margin towards the pleural surface in the context of pulmonary consolidation (Figure 2 and Figure 3).

Ultrasounds were able to penetrate in a consolidation, which is defined as lung “hepatization”. In the context of this consolidation, two types of air bronchogram have classically been described, both characterized by the presence of hyperechoic air in the context of a hypoechoic consolidation [22,23]. The dynamic air bronchogram is determined by the presence of air moving through the patent bronchi of the consolidation; in this case, the air is arranged in a branched manner within the consolidation, corresponding exactly to the bronchial branches, and in the dynamic phase it is possible to see the movement of the air itself inside the bronchi. The static bronchogram, on the other hand, is characteristic of resorptive atelectasis and in a smaller number of cases of pneumonia; in this case we see numerous air bubbles that do not change over time in the context of densification. Compared with cavitations, the air of the static bronchogram is represented by a greater number of air bubbles that appear smaller, irregular and arranged at numerous levels of depth. In cavitations, on the other hand, we see a single large air interface which represents the outermost air surface of the cavitation [22,23]. One of the advantages of this technique is that lung ultrasound is a “dynamic exam”: it is possible to evaluate real-time movement of the pleural surface and even the air within the bronchi in the air bronchogram, especially in the dynamic one [22]. It is known that LUS is a noninvasive and radiation-free approach, especially in young patients with suspected TB, and it has good sensitivity, together with CXR, compared with computed tomography exams [11]. In fact, LUS was originally used for detecting pneumonia in children, and this technique could also be used in daily practice in young patients with suspected or confirmed TB infection and in disease monitoring [16,22].

As already affirmed in our previous work, it is known that LUS and CXR can detect different findings in a complementary way: LUS can detect even small lesions at the pleural interface such as tiny septal and pleural thickening, subpleural micronodules and consolidations and pleural effusions [11]. On the other hand, CXR is a more panoramic and less sensitive exam in which small lesions appear more difficult to identify [24,25,26,27]. It is known that CXR is more sensitive for consolidations, nodules and cavitations, especially alterations that do not reach the pleural surface, and it showed a much lower sensitivity than CT (58%); however, its sensitivity was higher than the sensitivity demonstrated by LUS (33%) in our previous study [11]. Our results define a sensitivity percentage for LUS of 39.5% and for CXR of 68.4%, compared with CT; it could be interesting that by combining the sensitivity of LUS and CXR we can obtain a higher sensitivity value that could be sufficient in identifying lung and pleural alterations that could be caused by pulmonary TB. Moreover, in our study the LUS positive predictive value is 100%, meaning that once TB alterations are detected (with LUS, CT or CXR), LUS could be an optimal tool in the follow-up of peripheral lesions (such as cavitations) during antibiotic therapy.

Some authors suggest the use of a Low-Dose CT (LDCT) exam for the diagnosis of pleural–parenchymal alteration in TB. Nowadays, LDCT is used in different areas of thoracic pathology, such as for example in lung cancer screening protocols and in the evaluation of thoracic diseases in young patients and children (such as infections, asthma, congenital pathologies, trauma and ARDS) [28]. LDCT could be used as an initial examination in the evaluation of suspected pulmonary involvement of tuberculosis, especially in urgent cases or with suspected complications, for example, in cases of massive hemoptysis, suspected bleeding tuberculous caverns or cavitations, or bronchopleural fistulas and extensive alveolar hemorrhages. In cases of urgency and a hemodynamically unstable patient, the CT exam is the most suitable (with endovenous injection of a contrast medium) even in younger patients. In the case of a stable patient, with signs and symptoms attributable to a respiratory infection and a CXR suggestive of possible tuberculosis infection, LUS is the method of choice in nonurgent settings. In fact, as already mentioned, it does not involve exposure to ionizing radiation, is repeatable, is widely available in every hospital and has reasonable diagnostic accuracy, in particular if combined with CXR results.

This work has some limitations: This is a monocentric study with a limited number of patients. It also has to be remembered that LUS does not identify the deepest lesions that do not reach the pleural surfaces and it is an operator-dependent technique, based on the experience of the medical doctor. Another limitation is the difficulty of making a confident radiological diagnosis in the case of non-pulmonary TB in patients with similar clinical–radiological and epidemiological features; if needed, BAL or sputum analysis is fundamental in making a correct diagnosis [29,30]. On the other hand, our work underlines some advantages of LUS that are widely useful for radiologists and clinicians: it is widely available, is safer and cheaper compared with CT and can be easily performed in every hospital and health facility, in low- to middle-income countries as well [31,32]. Moreover, LUS does not require any patient transport because it is a bedside exam, nor does it require the presence of expensive CT scanners or dedicated technicians [33,34,35]. An interesting emerging field of study could be the introduction of new Artificial Intelligence (A.I.) techniques that can help in daily work and increase diagnostic performance, especially in “new user” radiologists [36]. This concept could be useful especially in medical centers in low- to middle-income countries where training resources are few; applying A.I. tools, accuracy, examination times and confidence could also be improved in the ultrasound field.

## 5. Conclusions

In conclusion, LUS could be a valid tool in identifying pulmonary parenchymal TB lesions, such as cavitations, with a relative low sensitivity but a very high positive predictive value. Combining LUS with CXR allows improvement of global sensibility and this is crucial especially in high-burden/low CT availability settings or if the patient is not suitable for CT itself. Radiologists have to remember that differential diagnosis between pulmonary TB and other lung diseases is also difficult with CT exams in some cases, but subpleural cavitations or other parenchymal alterations detected with LUS can be followed up without radiation exposure, allowing confident radiological disease monitoring especially during and after medical therapy. Unfortunately, nowadays many studies are limited case series or single-center experiences; in the near future we hope to have multicenter studies available to validate the many advantages of LUS investigation, particularly in this specific group of patients, which extends throughout the world.

## Figures and Tables

**Figure 1 diagnostics-14-00522-f001:**
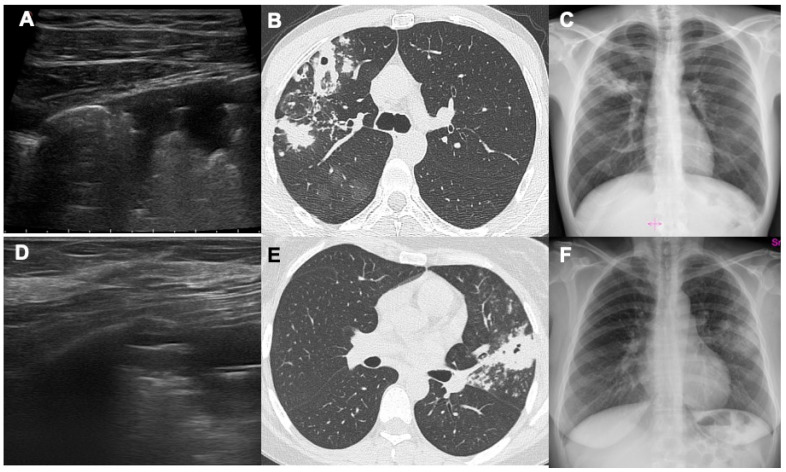
This figure shows two cases ((**A**–**C**) and (**D**–**F**)) where LUS, CT and CXR were compared. In both cases there are two cavitations, one in the right upper lobe (first case) and one in the left upper lobe (second case), and both alterations reach the pleural surfaces. Combining LUS and CXR, it is possible to define the disease extension and detect the main alterations, sufficient for a suspected TB diagnosis.

**Figure 2 diagnostics-14-00522-f002:**
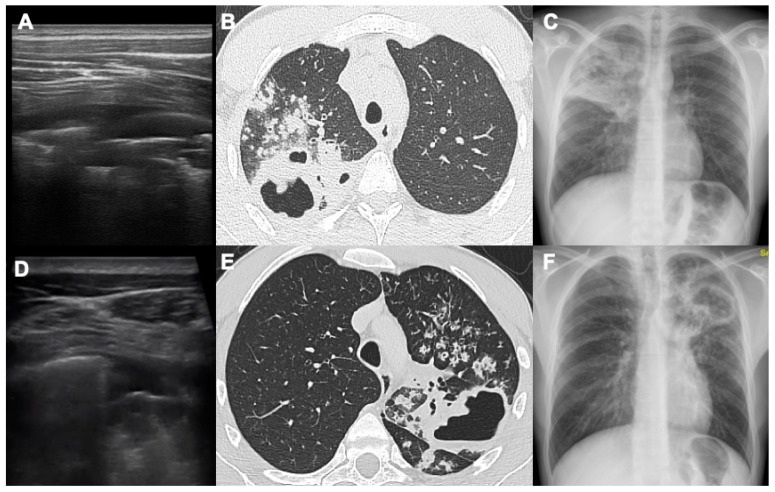
This figure shows two cases ((**A**–**C**) and (**D**–**F**)) where LUS, CT and CXR were compared. In both cases, there are two extensive cavitations, one in the right upper lobe (first case) and one in the left upper lobe (second case), and both alterations reach the pleural surfaces. In these two cases, lung involvement of the upper lobes is important and CXRs clearly demonstrate it; of course, the CT exams in B and E show better the complete parenchymal involvement around the two cavitations.

**Figure 3 diagnostics-14-00522-f003:**
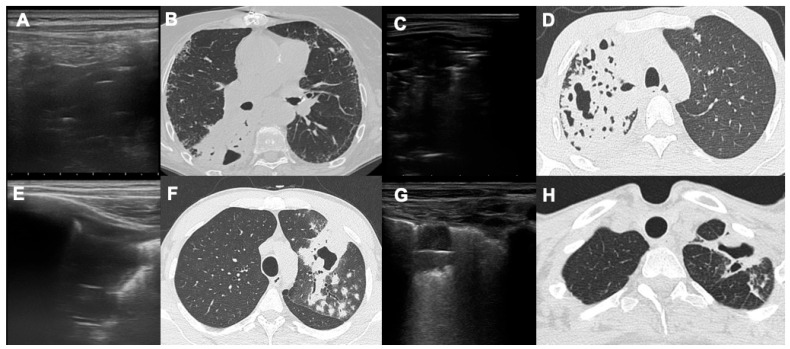
This figure shows four cases of pulmonary cavitations with extensive areas of consolidations around them (cases (**A**,**B**); (**C**,**D**); (**E**,**F**); and (**G**,**H**)); images in (**A**,**C**,**E**,**G**) are LUS scans demonstrating the presence of hypoechoic areas below the pleural surface, with static or dynamic bronchograms, which are the linear hyperechogenic lines within the consolidations, parallel with the pleura.

**Table 1 diagnostics-14-00522-t001:** Patients’ characteristics and differential diagnoses in non-pulmonary TB subjects. HIV: human immunodeficiency virus; COPD: chronic obstructive pulmonary disease; TB: tuberculosis.

	Pulmonary TB	Non-Pulmonary TB	Total	*p*-Values
**Total (%)**	58 (70.7)	24 (29.3)	82 (100)	
**Males (%)**	42 (68.9)	9 (37.5)	51 (62.2)	0.131
**Origin (%)**				
*Italy*	18 (31)	10 (41.6)	28 (34.1)	0.523
*Other countries*	40 (69)	14 (58.4)	54 (65.9)	0.670
**Risk Factors (%)**				
*Smoking*	20 (34.5)	10 (41.6)	30 (36.6)	0.678
*HIV*	1 (1.7)	0 (0)	1 (1.2)	
*COPD*	9 (15.5)	3 (12.5)	12 (14.6)	0.760
*Active cancer*	2 (3.4)	2 (8.3)	4 (4.9)	0.378
*Immunosuppression*	8 (13.7)	3 (12.5)	11 (13.4)	0.891
*TB contact*	3 (5.2)	0 (0)	3 (3.6)	
*Previous TB*	8 (13.7)	4 (16.6)	12 (14.6)	0.773
*Homeless*	2 (3.4)	1 (4.1)	3 (3.6)	0.879

**Table 2 diagnostics-14-00522-t002:** Patients’ differential diagnoses in non-pulmonary TB subjects. COPD: chronic obstructive pulmonary disease; TB: tuberculosis.

Other Diagnosis	Patients (%)
*Pneumonia (bacterial)*	11 (45.8)
*COPD*	4 (16.6)
*Previous TB*	3 (12.5)
*Pneumonia (viral)*	2 (8.3)
*Lung cancer*	2 (8.3)
*Organizing pneumonia*	2 (8.3)
** *Total* **	**24** (100%)

**Table 3 diagnostics-14-00522-t003:** Number of patients with cavitations among pulmonary TB and non-pulmonary TB patients detected with each diagnostic imaging tool. TB: tuberculosis.

	Pulmonary TB	Non-Pulmonary TB	*p*-Values
** *Computed tomography* **	32/58 (55.2%)	6/24 (25%)	0.112
** *Chest X-ray* **	23/58 (40%)	4/24 (16.7%)	0.135
** *Lung ultrasound* **	11/58 (19%)	4/24 (16.7%)	0.838

## Data Availability

Data is unavailable due to privacy restrictions.

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
