# Peer review of "Parenchymal Cavitations in Pulmonary Tuberculosis: Comparison between Lung Ultrasound, Chest X-ray and Computed Tomography"

_diagnostics, 2024, doi:10.3390/diagnostics14050522_

Round 1

Reviewer 1 Report

Comments and Suggestions for Authors

This study is prospective single center on the detection of lung cavitations using lung ultrasound (LUS) in patients with pulmonary tuberculosis (TB) compared with chest computed tomography (CT) and chest x-ray, to describe its diagnostic accuracy. Authors demonstrated that LUS sensitivity was 39.5%, specificity 100%, PPV 100%, NPV 65.7%. No false positive cases were found. This study may provide some useful information on the diagnostic role of LUS in patients with cavitary pulmonary TB. I have some comments.

Major comments

1. Ultrasound is a test which results can vary greatly depending on the operator's skill level. How many radiologists attended in this study? And how skilled are radiologists? Please describe it in “Materials and Methods” session.

2. Was there a difference in detection rate of LUS depending on the cavitation size?

What size of the lesions and depth from pleura could be to detect it with the ultrasound?

Minor comments

1. In line 24, Please describe the full term of the abbreviation, “BAL.

2. In line 36, World Health Organization

Please corrected to World Health Organization (WHO).

3. In line 39, World Health Organization (WHO)

Please corrected to WHO.

4. In line 65, Computed Tomography

Please corrected to CT.

5. In line 75, Computed Tomography

Please corrected to CT.

6. In line 76, X-ray

Please corrected to CXR.

7. In line 79-80, chest computed Tomography and chest x-ray

Please corrected to chest CT and CXR.

8. In line 92, computed tomography and chest X-ray

Please corrected to CT and CXR.

9. In line 144, Chest X-Ray

Please corrected to CXR.

10. In line 133-134 (Table 1), Please describe the full term of the abbreviation, “HIV.

Comments on the Quality of English Language

Minor editing of English language required.

Author Response

Dear Reviewer,

thank you for the revision. We have made all the  changes and additions you suggested (especially in the M&M section).

Best regards,

All authors

Reviewer 2 Report

Comments and Suggestions for Authors

I read with attention the article on the use of ultrasound, radiological examination, and computed tomography of the chest in the assessment of tuberculosis cavities

My comments concern several issues:

- Table No. 1 should contain significance calculations (p-level)

- Table No. 3 should contain significance calculations (p-level). After calculating the significance in this table, it turns out that CT diagnoses cavities in TB significantly better than in non-TB patients, and the LUS shows these changes similarly (similarly badly).

- What I missed in the discussion was the paragraph about low-dose CT and the possibility of using this technique to diagnose lung cavities.

- The possibilities of diagnosing cavities using LUS due to the method must be limited. Originally, lung ultrasound was used in the diagnosis of pneumonia in children, so can we risk that the diagnosis of cavernous lesions may also be more beneficial in children? I suggest adding this information to the discussion.

- Can we think about using LUS to assess the remission of lesions in treated patients in whom cavernous lesions are visible? Maybe this is an idea for further research?

Author Response

Dear Reviewer,

thanks for the interesting comments, we have included all the suggested ideas in the text (in the discussion paragraph). We also calculated and added the p-values in Tables 1 and 3.

Best regards,

all Authors

Reviewer 3 Report

Comments and Suggestions for Authors

A growing and fascinating area of ultrasound imaging is lung ultrasound (LUS). The potential benefits of an effective ultrasound method focused on the monitoring and diagnosis of lung disease, despite the difficulties of imaging an organ largely filled with air, provide a tremendous stimulus for research in this area. This study was designed to evaluate the potential of lung ultrasound in recognizing lung cavitations, compared with chest computed tomography and chest x-ray. The work has been thorough and in compliance with all the necessary validation criteria. The authors conclude that LUS, with a relatively low sensitivity but a very high positive predictive value, is a valid tool for identifying pulmonary TB lesions such as cavitations. However, careful consideration should be given to the potential risks associated with the use of unnecessary high pressure fields in lung examinations. In fact, the presence of air in the lungs would seem to suggest the use of an MI that is well below the limit of 1.9 set for standard soft tissue ultrasound imaging. The ability to obtain high quality LUS data is not in conflict with this. In fact, lower MIs are sufficient for imaging because the pleural line is only a few centimetres from the skin.

Author Response

Dear Reviewer,

Thank you very much for your comments on our work. We agree on everything, even on the fact that radiologist or physicians should not apply greater pressure in the chest ultrasound study (as sometimes it is needed during an abdominal US).  We have included this concept in the Materials and Methods section. We thank you again for your work,

Best regards,

all Authors